# Effectiveness of Paliperidone Palmitate in Reducing Acute Psychiatric Service Use for Patients Suffering from Psychosis—A Retrospective Mirror-Image Study

**DOI:** 10.3390/ijerph20043403

**Published:** 2023-02-15

**Authors:** Chun Ting Chan, Swapna Verma, Mythily Subramaniam, Edimansyah Abdin, Jenny Tay

**Affiliations:** 1Institute of Mental Health, Singapore 539747, Singapore; 2Duke-NUS Medical School, Singapore 169857, Singapore

**Keywords:** psychosis, long-acting antipsychotic, paliperidone palmitate, outcome, mirror-image study

## Abstract

Poor adherence to antipsychotic treatment is a leading cause of relapse for patients suffering from psychotic disorders and the use of long-acting injectable antipsychotics (LAI) may lead to improved clinical outcomes. This was a 1-year mirror-image study examining the clinical outcomes after monthly administration of paliperidone palmitate (PP1M). The primary outcome measure was the total days of psychiatric hospitalization in the 1-year before and 1-year after initiation of PP1M. Data from 158 patients were included in the study. Most of the patients suffered from schizophrenia. In the year after initiation of PP1M, the mean number of hospitalization days fell from 106.53 to 19.10 (*p* < 0.001). There were significant reductions in the mean number of hospitalizations and emergency room visits. The use of paliperidone palmitate is associated with significant reduction in both the number of admissions and days of psychiatric hospitalization.

## 1. Introduction

Schizophrenia and other psychotic disorders are often chronic in nature and long-term maintenance treatment is effective in reducing the risk of relapse and rehospitalization. Suboptimal adherence to antipsychotic treatment is common in patients suffering from psychosis [1,2] and is one of the main causes of relapse [3,4]. Non-adherence to treatment leads to poorer clinical and functional recovery outcomes [5], while maintenance treatment was found to be effective in protecting patients from environmental stress and successfully engaging them in social interventions and rehabilitation programs.

It has been estimated that approximately 40% of patients stop taking their antipsychotic treatment within 1 year, and 75% stop within 2 years [1]. Factors contributing to treatment non-adherence include poor insight into their illness and the need for treatment, side-effects of treatment, severity of psychotic symptoms and cognitive impairment [6,7]. Strategies to improve adherence include educating patients on their psychiatric disorder and treatment, motivational interviewing, financial incentives and the use of long-acting injectable antipsychotics (LAI) [6,8].

There are several advantages of LAIs as compared to oral antipsychotics [9,10]. LAIs bypass the effect of first-pass metabolism in the liver and offer a more consistent bioavailability and more predictable correlations between dosage and plasma drug level. The administration of LAIs every few weeks eliminates the need for patients to take antipsychotic tablets daily, reducing the risk of covert non-adherence to antipsychotic treatment and overdose [11,12]. Patients who have been prescribed second-generation antipsychotic LAIs remain on treatment for longer compared to those who have been prescribed second-generation oral antipsychotics, first-generation antipsychotics and first-generation antipsychotic LAIs [13]. In addition, the improvement in the control of their psychiatric disorder associated with LAIs allow the patients to focus on recovery and maximize their potential to live a satisfying and purposeful life.

However, there have been inconsistent findings in the literature over the effectiveness of long-acting injectable versus oral antipsychotics in preventing relapse. A systematic review by Schneider et al. did not find clear differences between antipsychotics, regardless of the route of administration, in preventing the relapse of schizophrenia [14]. One reason for the inconsistency is the choice of study design [15]. Randomized controlled trials (RCT) are considered the gold standard in identifying differences in treatment efficacy as they avoid many of the issues associated with confounders that affect results from observational studies. However, compared to real-life settings, participants in RCTs are scrutinized and checked for their adherence to treatment and the knowledge that their treatment adherence is being observed may affect their behavior and improve treatment adherence. These biases may have led to the observation in RCTs that the benefits of LAIs are not significantly superior to those of oral antipsychotics [16]. In contrast, observational studies are more similar to real-life clinical settings and studies utilizing observational designs tend to confer an advantage to LAIs [15,16]; however, such designs may not be able to control for selection bias and other confounding factors. These factors lead to challenges in comparing results from studies employing different study designs.

To mitigate issues associated with selection bias in observational studies, observational mirror-image studies have been conducted. Mirror-image designs are studies where outcomes are compared before and after a change in medication. The major advantage of such a design is that it allows for study participants to act as their own control. A meta-analysis of mirror-image studies, where the outcomes between periods of oral antipsychotic and LAI treatments of the same patients are compared, showed that the use of LAIs significantly reduced the risk of hospitalization [15,17].

Compared to first-generation antipsychotics, second-generation antipsychotics are associated with a lower risk of developing extrapyramidal motor and sexual side-effects [10,18]. This may lead to improved adherence to treatment and, consequently, reduced risk of relapse and acute psychiatric service use. Studies have found that second-generation antipsychotics are favored by patients even though the efficacy in controlling positive symptoms is similar to first-generation antipsychotics [19]. An example of a second-generation antipsychotic is paliperidone, which is a 9-hydroxy metabolite of risperidone, a selective dopamine (D2) and 5-hydroxytryptamine 2A (5-HT2A) antagonist. Paliperidone palmitate is a long-acting injectable antipsychotic available in Singapore since January 2012. Its advantages include a rapid onset of action, and it is available in once-a-month (Paliperidone Palmitate Once a Month—PP1M) or once-in-3-months’ preparations.

We aimed to evaluate whether the use of PP1M was associated with a reduction in acute psychiatric service use in patients who had been on other preparations of oral antipsychotic or LAI. The primary outcome was total days of psychiatric hospitalization in the 12 months before and 12 months after initiation of PP1M. We hypothesized that patients prescribed PP1M would have a lower number of hospitalization days in the year after it was first administered.

## 2. Methods

### 2.1. Study Design

The study employed anonymized data collected from the Institute of Mental Health (IMH) in Singapore. The IMH is the only tertiary psychiatric hospital in Singapore offering both inpatient and outpatient psychiatric care to people of all ages. The IMH serves the population of Singapore which comprises 5.70 million people.

A retrospective mirror-image study was designed to determine the effectiveness of PP1M compared to oral antipsychotic medication or another LAI (see Figure 1). We defined the date of the first injection of PP1M as the index event and used 4 days after this date as the cut-off point for determining the pre-initiation and post-initiation periods.

The decision regarding the choice of medication, as well as the route of administration, was made by the attending psychiatrists and their patients. This was an observational study where participation in the study would not have influenced the choice of treatment for the participating patients.

De-identified data were extracted from the hospital electronic medical records for inpatients and outpatients who were administered PP1M between 1 January 2017 and 31 December 2018. Participants did not have to attend additional visits for the study, and statistical analyses were conducted on de-identified data only.

A waiver of informed consent was granted for this study. The study design was approved by the Clinical Research Committee of the IMH and the Domain Specific Review Board of the National Healthcare Group.

### 2.2. Study Population

Patients who were prescribed oral antipsychotics for 12 months, followed by the initiation of treatment with PP1M between 1 January 2017 and 31 December 2018, and remained on PP1M for at least 3 months were selected from the hospital database. Inclusion criteria included patients between the age of 21 and 65 years; patients with the Diagnostic and Statistical Manual of Mental Disorders, Fourth Edition (DSM-IV) diagnoses of schizophrenia, schizophreniform disorder, brief psychotic disorder, schizoaffective disorder, delusional disorder, psychotic disorder not otherwise specified, psychotic disorder due to a general medical condition and substance-induced psychotic disorder; and who were on PP1M for at least 3 months. Patients who were prescribed Clozapine in the past and long-stay inpatients who were admitted for more than 3 months were excluded.

A Consolidated Standards of Reporting Trials (CONSORT) flow diagram of the cohort is shown in Figure 2.

### 2.3. Study Measures

Patient data including age, gender, ethnicity, marital status, employment status, diagnoses and pharmacological treatments were obtained for all potential participants from the electronic records. The primary outcome was total hospitalization days. The secondary outcomes were number of psychiatric admissions, number of IMH emergency room visits and number of visits by community mental health teams. The effectiveness of PP1M in reducing acute psychiatric service use was determined by comparing the outcome measures in the pre-initiation period to the post-initiation period. Data were collected for 365 days before and after introduction of PP1M.

In mirror-image studies, the location of the mirror point may significantly influence the study results. For both inpatients and outpatients, we decided to set the mirror point as 4 days after the initiation of PP1M based on findings from the studies by Alphs et al. and Li at al. [20,21], which demonstrated significant improvement in the Positive and Negative Syndrome Scale (PANSS) total score 4 days after the first PP1M injection (*p* < 0.05). For inpatients, this assigns the first part of the index admission to previous treatment, and the second part to PP1M.

The primary outcome measure was calculated as total days of psychiatric hospitalization in the 12 months before and 12 months after initiation of PP1M, and this was obtained from the hospital database.

### 2.4. Statistical Analysis

The required sample size was estimated using a paired test comparing two correlated means with a two-sided significance level of 5% and power of 90%. The sample size calculation was based on the study by Taylor et al. [22] and was used to compare the number of hospital admissions and bed days before and after PP-LAI using the ‘mirror-image’ method. In order to replicate their findings, we required at least 123 patients. To account for approximately 40% missing or incomplete data, we needed approximately 172 patients to be able to reject the null hypothesis that the population means of the two groups were equal.

All statistical analyses were performed using Statistical Product and Service Solutions (SPSS) version 18 for Windows. Research team members involved in the data extraction process were not involved in data analysis. Sociodemographic and clinical characteristics of patients were summarized using descriptive statistics. Means, standard deviations (SDs) and Inter-Quartile Range (IQR) were calculated for continuous data and frequencies and percentages were calculated for categorical data. The continuous data were tested for normal distribution using histograms and the Shapiro–Wilk test of normality. Since the continuous data were not normally distributed, comparisons before and after PP1M were performed using the non-parametric Wilcoxon signed-rank test for paired data and the bootstrap paired-sample *t*-test. The *t*-test is a statistical method used to test the hypothesis that two paired samples have the same mean values. We performed subgroup analyses on patients who received oral antipsychotics versus those on PP1M. All statistically significant tests were set at a *p* value < 0.05.

## 3. Results

A total of 863 patients were prescribed PP1M during the study period. In total, 158 patients fulfilled the inclusion criteria of the study, and their data were included for analysis. Their demographic details are presented in Table 1. The mean age of the study population was 46.28 years (SD = 11.33), and 50.63% were male. The most common diagnosis was schizophrenia, comprising 82.9% of the sample. The mean dose of PP1M prescribed was 103.13 (SD = 31.79). The common antipsychotic medications that were prescribed in the pre-initiation period were Risperidone (35.5%), Quetiapine (8.8%) and oral Paliperidone (8.1%).

Results were reported for the entire cohort of 158 patients, and sub-analyses were conducted for those who were administered PP1M during an inpatient stay (*n* = 57) and those who received their first PP1M injection as an outpatient (*n* = 101).

### 3.1. Mean Total Length of Hospitalization

The mean length of stay in the year before starting PP1M was 106.53 days (SD = 127.44) which decreased significantly to 19.10 (SD = 53.86) in the year after (*p* < 0.001).

### 3.2. Mean Number of Hospitalization

The mean number of hospitalizations in the year before starting PP1M was 1.47 (SD = 1.03) which reduced significantly to 0.49 (SD = 0.98) in the year after (*p* < 0.001).

### 3.3. Mean Number of Emergency Room Visits

The mean number of emergency room visits in the year before starting PP1M was 1.77 (SD = 1.75) which decreased significantly to 0.73 (SD = 1.46) in the year after (*p* < 0.001).

### 3.4. Mean Number of Visits by the Community Mental Health Team (CMHT)

The mean number of CMHT visits in the year before starting PP1M was 1.42 (SD = 3.78) and 1.60 (SD = 3.80) in the year after. There was no significant difference between the two observation periods (*p* = 0.8145).

There were no significant differences in age, gender, race, marital status and diagnosis between the groups who were administered PP1M in the outpatient or inpatient settings.

The results are summarized in Table 2.

## 4. Discussion

Data from this study demonstrated the effectiveness of PP1M in reducing the number of admissions and the number of days of hospitalization in the year after it was initiated in patients with schizophrenia-spectrum disorders being treated in a tertiary psychiatric hospital. The reductions can be attributed to improved control of the illness due to improvement in treatment adherence from the use of PP1M.

We also found a reduction in the mean number of psychiatric emergency room visits post-PP1M initiation. This implies that patients on PP1M were stable, thus negating the need for unscheduled visits to the emergency room.

This overall improvement in mental wellness would enable patients to live life as they want to and to achieve their full potential. They would be able to engage themselves in activities that are meaningful to them and build and maintain relationships with people around them.

There was no significant difference in the number of CMHT visits pre- and post-PP1M. CMHT provides community-based care in a multi-disciplinary setting. The services provided by CMHT include liaising with primary and specialist health care providers, offering acute and maintenance treatment in the community, as well as offering rehabilitative services to those experiencing disabilities. In view of the range of services offered to patients during different phases of illness, the frequency of the visits may not reflect the mental status of the patients. Our results are consistent with findings from previous studies involving PP1M. In a naturalistic study involving 114 patients suffering from psychotic disorders [23], the authors found that the use of PP1M reduced the mean annual hospitalization days from 45.8 to 38.5 days, while decreasing the mean number of admissions from 1.9 to 0.64. Before the administration of PP1M, the participants’ mean medication-possession ratio was 43%. This suggests that the patients under study were only partially adherent to the prescribed oral antipsychotic treatment. The MPR, however, was in line with results from cross-sectional surveys of outpatients suffering from schizophrenia [1,2,24].

Oh et al. conducted a retrospective observational mirror-image study which included data of 46 patients from South Korea [25]. The demographics of the study population were similar to our present study and, as in our study, most of the participants suffered from schizophrenia. The authors similarly used a retrospective mirror-image study design with 12 months of observation before and after PP1M administration. The study demonstrated a significant reduction in the mean number of admissions from 0.83 to 0.17 one year after the initiation of PP1M, while the mean number of total bed days decreased from 24.85 to 8.74.

In another study involving a longer observation period, Taylor et al. reported a 4-year observational mirror-image study involving 225 patients who were prescribed PP1M [22]. They found a significant reduction in the mean number of admissions and mean hospitalization days in the 2 years after PP1M administration. The differences remained significant, regardless of whether the index admissions (where PP1M was administered) were included or excluded from analysis.

In our study, the number of days of hospitalization pre-PP1M was higher than in other published mirror-image studies [22,23,25]. Several factors that are known to affect the length of stay, including more severe illness, the need for inpatient rehabilitation and discharge to rehabilitation facilities [26]. Being the only tertiary psychiatric hospital in Singapore, many patients who are managed at the IMH suffer from more severe illnesses, requiring more intensive and a longer duration of care. The IMH offers an inpatient rehabilitation program spanning 8 weeks and there is often a waiting list for patients to be discharged to community rehabilitative facilities. These factors may have contributed to the higher number of hospitalization days found in our study.

Our study did not compare the clinical efficacy between PP1M and other LAIs, and evidence from the current literature does not suggest that PP1M enjoys a higher efficacy compared to other first- and second-generation LAIs. When comparing PP1M and Haloperidol Decanoate, McEnvoy et al. did not find any statistically significant difference in treatment failure in patients with Schizophrenia and Schizoaffective disorder [27]. In a randomized controlled trial comparing PP1M and Risperidone long-acting injectable, the authors demonstrated similar efficacies in the two LAIs [28]. In a naturalistic study comparing the clinical and functional outcomes of participants who had been prescribed PP1M, Aripiprazole Prolonged Release and Haloperidol Decanoate, participants across the three groups showed similar and significant reductions in urgent psychiatric consultations and hospitalizations [29].

The major advantage of this mirror-image study is the utilization of a naturalistic design, allowing for the observation of outcomes in a real-world clinical setting.

The study has several limitations. First, the study subjects were restricted to those who had been on PP1M for at least 3 months, and this may have led to selection bias; thus, patients who did not have a favorable response to PP1M during the first 3 months would have been excluded. Second, the reasons for admission were not collected and the admissions were assumed to be for mental health reasons. Admissions due to psychosocial reasons or the management of medication side-effects could not be excluded. Third, we inferred that the results observed were due to the use of PP1M and we did not exclude the possibility of the effect of other antipsychotic and non-antipsychotic medications on the findings. Fourth, we did not utilize other methods of evaluating the therapeutic effects in patients suffering from a psychotic disorder (such as rating scales to assess the severity of psychopathology) and relied solely on indirect outcomes such as the duration and number of psychiatric admissions. Finally, variables that could account for disparity in outcomes, such as duration of illness, premorbid level of functioning and mode of illness onset, were not available for analysis.

## 5. Conclusions

The findings from our study support the use of PP1M in reducing the number of psychiatric admissions, psychiatric inpatient days and acute emergency psychiatric services use in an Asian population. Future studies involving longer observational periods both before and after PP1M administration should be considered to evaluate the long-term effectiveness and cost efficacy of PP1M.

## Figures and Tables

**Figure 1 ijerph-20-03403-f001:**
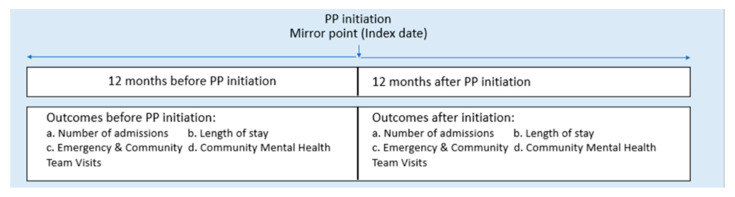
A mirror-image study design.

**Figure 2 ijerph-20-03403-f002:**
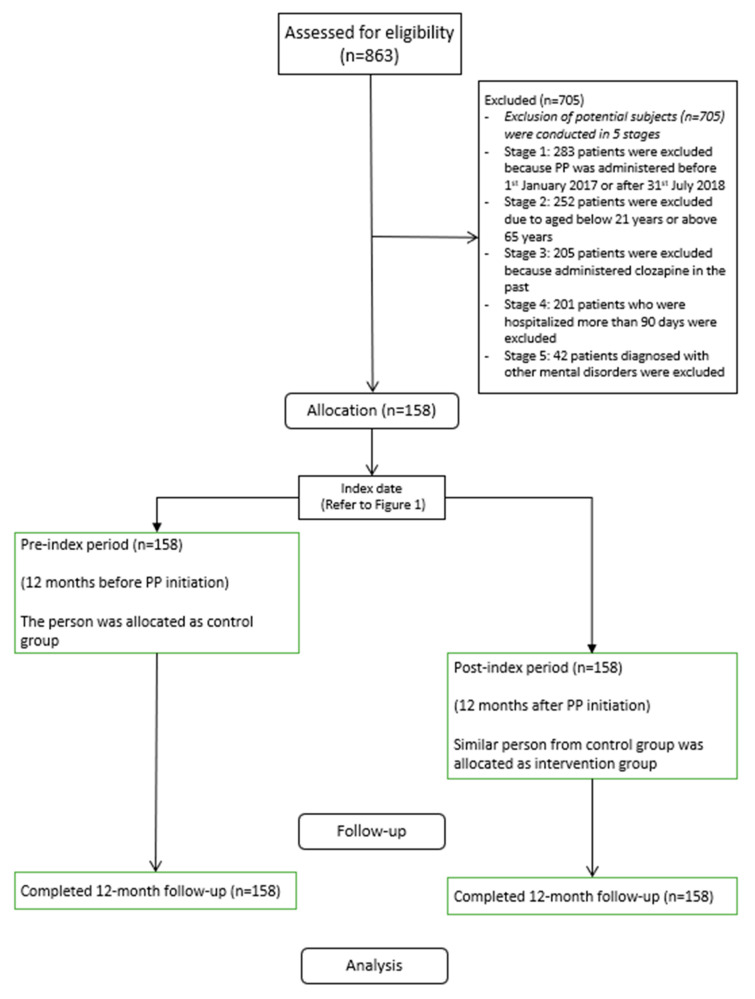
Modified CONSORT flow diagram for a mirror-image study.

**Table 1 ijerph-20-03403-t001:** Sociodemographic characteristics of patients with schizophrenia spectrum disorders (*n* = 158).

	*n*	%		
Gender				
Female	78	49.37		
Male	80	50.63		
Ethnicity				
Chinese	113	71.52		
Indian	16	10.13		
Malay	20	12.66		
Others	9	5.7		
Marital status				
Single	82	51.9		
Married	29	18.35		
Divorced	5	3.16		
Unknown	42	26.58		
Employment				
Employed	2	1.27		
Student	4	2.53		
Unemployed	6	3.8		
homemaker	2	1.27		
Unknown	144	91.14		
Deceased				
No	155	98.1		
Yes	3	1.9		
	Mean	SD	Min	Max
Age	46.28	11.33	25	66
Number of injections at Post-PP1M injection (Median = 4)	6.09	5.18	1	29

**Table 2 ijerph-20-03403-t002:** Hospital utilization before and after PP1M injection initiation among everyone (*n* = 158): inpatient (*n* = 57) and outpatient (*n* = 101).

	Pre-Injection				Post-Injection					
	Mean	Median	Range		Mean	Median	Range		Bootstrap	Wilcoxon
	(SD)	(IQR)	(Min)	(Max)	(SD)	(IQR)	(Min)	(Max)	Paired	Signed
									*t*-Test	Rank
										Test
Total of Length of Stay (days)	106.53 (127.44)	25.5 (210)	0	419	19.10 (53.86)	0 (10)	0	337	<0.001	<0.001
Number of Psychiatric Admissions	1.47 (1.03)	1 (1)	0	5	0.49 (0.98)	0 (1)	0	6	0.001	<0.001
Inpatient	1.05 (0.23)	1 (0)	1	2	0.21 (0.41)	1 (0)	0	1	<0.001	<0.001
Outpatient	1.71 (1.21)	2 (1)	0	5	0.65 (1.16)	0 (1)	0	6	<0.001	<0.001
Number of Emergency Visits	1.77 (1.75)	1 (1)	0	10	0.73 (1.46)	0 (1)	0	9	0.001	<0.001
Inpatient	1.19 (0.61)	1 (0)	0	4	0.33 (0.72)	0 (0)	0	4	<0.001	<0.001
Outpatient	2.10 (2.08)	2 (2)	2	10	0.95 (1.71)	0 (1)	0	9	<0.001	<0.001
Number of CMHT visits	1.42 (3.78)	6 (8)	0	21	1.60 (3.80)	6.5 (8)	0	20	0.579	0.8145
Inpatient	0.89 (2.53)	0 (0)	0	11	1.37 (3.58)	0 (0)	0	20	0.335	0.9807
Outpatient	1.72 (4.31)	0 (0)	0	21	1.73 (3.93)	0 (0)	0	15	0.430	0.688

## Data Availability

The data presented in this study are available on request from the corresponding author. The data are not publicly available due to privacy.

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
