# Peer review of "Effectiveness of Paliperidone Palmitate in Reducing Acute Psychiatric Service Use for Patients Suffering from Psychosis—A Retrospective Mirror-Image Study"

_ijerph, 2023, doi:10.3390/ijerph20043403_

Round 1
Reviewer 1 Report
Overall, the study is interesting, and we need more research to provide appropriate care for people with serious mental illness. However, there is room for improvement. The author should address how this study adds to knowledge.
The authors identified specific problems, but they are not well supported be reference. The reference is quite old. The effectiveness of long-acting injectables on patient outcomes has been well documented; therefore, the authors should argue the rationale for the study, which is not well supported by the reference. The author should justify the use of Paliperidone palmitate (every month). What is the new information/knowledge from this study should be addressed.
In the inclusion criteria, the diagnosis is quite heterogeneous, such as schizophrenia vs. psychotic disorder due to a general medical condition or substance-induced psychotic disorder. The authors may focus on schizophrenia and schizoaffective disorder.
The age range is also wide (21 to 65 years old). In general, patients get older, their psychiatric symptoms get stable. This can impact the result.
The inclusion/exclusion criteria do not address whether inpatients or outpatients are included. However, in Table 2, the authors compute the total length of stay based on inpatients and outpatients, which influence the results, such as an ER visit or hospitalization. Another outcome, emergent visits, is not addressed clearly in terms of measurement.
This study used a mirror-image design, but the study did not use the same group comparison based on the information from Figure 2. The authors mentioned that similar people from the control group were assigned to the intervention group, but what they mean by "similar" is unclear.
Confounding variables such as the length of illness, adherence to the oral medications, and other psychotropic medications taken are not controlled. The result is also subject to potential time or cohort effects due to methodological issues, which should be addressed in the limitation.
As mentioned by the authors, PP1M is available once a month, but the max number of injection at Post-PP1M was 29 (Table 1).
There was statistic with an asterisk in Table 1, but no footnote.
Author Response
Dear Reviewer,
Thank you for the comments. Please kindly refer to the attached word file for our response.

Reviewer 2 Report
Journal: I would recommend instead the submission to Journal of Psychiatry, not Public Health. As the article brings nothing new to the international scientific literature, I would rather recommend submission in a regional journal of Psychiatry, such as any other journal base in the Australasian or Eastern Asia: Australia, New Zealand, Singapore, Hong Kong, Macau, Taiwan, China, Korea, Japan, etc…
First Page: I did not find any affiliation or Correspondence contact for any authors.
Line 26: please, delete “Treatment adherence and long-term outcomes”, for a better citation.
Line 34: please avoid non-scientific expressions such as “illness”. Use instead “psychiatric disorder”.
Line 47: please avoid non-scientific expressions such as “mental well-being”. Use instead “psychiatric disorder”.
Line 45: the authors should also mention the “third generation antiopsychotics” (eg aripiprazole, etc).
Line 85: please, explain the meaning of the acronym “PP1M”.
Figure 1: please, do not use color, for ecological and economic reasons.
Line 82: please, explain the meaning of the acronym “5-HT2A”.
Line 119: please, explain the meaning of the acronym “DSM-IV”. Please provide equivalent diagnostic codes for ICD-11. Please provide complementary exams done to all patients, eg: how many did EEG to exclude temporal lobe epilepsy, how many did MRI brain scan to exclude CNS tumors, how many did lumbar puncture to exclude auto-immune encephalitis, how many did bloodwork to exclude deficit of vitamins or hormonal disease, how many did urinalysis to exclude drug induced psychoses, how many did psychological assessment to discard personality disorder or mental retardation, how many did neuropsychological assessment to discard dementia, etc… Please introduce a new paragraph regarding concepts such as secondary schizophrenias, pseudo-schizophrenias, symptomatic schizophrenias or schizophrenia-like psychosis. Acknowledge that there are many imitators of schizophrenia in contemporaneous medicine. Schizophrenia is, ideed, the great imitated of psychiatry. Authors will ask about the prevalence of the specific organic psychoses in your sample (how many tumors, how many epilepsies, how many dementias, how many cocaine addicts, etc)?. How many patients were medicated with off-label paliperidone treatment? Pease provide the mean chlorpromazine and/or olanzapine equivalent for all patients. Please quantifyc the body mass index, pack per year abuse of nicotine and cafeíne / tea daily abuse.
Line 126: please, explain the meaning of the acronym “CONSORT”.
Figure 2: please, do not use color, for ecological and economic reasons. This image has no quality for publication. Please, explain the meaning of the acronyms / initials “PP” and “n”.
Line 142: please, explain the meaning of the acronym “PANSS”.
Line 157: please, explain the meaning of the acronym “SPSS”.
Line 165: please, explain the meaning of the acronym “t-test”.
Table 1: try to avoid the division of one table in two pages. Please explain the meaning of “3*” after “Yes” for deceased. The Readers, will naturally, ask for the cause of death of those three patients.
Line 191: pleas capitalize the first letter for every word that contributes to the acronyms “CMHT”.
Table 2: please, explain the meaning of all acronyms, such as “SD”, “IQR”, etc…
Line 311: please remove “feb” from Reference “11”, please remove “jun” from Reference “24”, etc… I never saw this format of citation, eg (Year month). Weird?
